# Encouraging and Detecting Preferential Incipient Slip for Use in Slip Prevention in Robot-Assisted Surgery

**DOI:** 10.3390/s22207956

**Published:** 2022-10-19

**Authors:** Ian Waters, Dominic Jones, Ali Alazmani, Peter Culmer

**Affiliations:** 1School of Mechanical Engineering, University of Leeds, Leeds LS2 9JT, UK; 2School of Electronic and Electrical Engineering, University of Leeds, Leeds LS2 9JT, UK

**Keywords:** surgical robotics, grasping, slip, force, tactile sensing

## Abstract

Robotic surgical platforms have helped to improve minimally invasive surgery; however, limitations in their force feedback and force control can result in undesirable tissue trauma or tissue slip events. In this paper, we investigate a sensing method for the early detection of slip events when grasping soft tissues, which would allow surgical robots to take mitigating action to prevent tissue slip and maintain stable grasp control while minimising the applied gripping force, reducing the probability of trauma. The developed sensing concept utilises a curved grasper face to create areas of high and low normal, and thus frictional, force. In the areas of low normal force, there is a higher probability that the grasper face will slip against the tissue. If the grasper face is separated into a series of independent movable islands, then by tracking their displacement it will be possible to identify when the areas of low normal force first start to slip while the remainder of the tissue is still held securely. The system was evaluated through the simulated grasping and retraction of tissue under conditions representative of surgical practice using silicone tissue simulants and porcine liver samples. It was able to successfully detect slip before gross slip occurred with a 100% and 77% success rate for the tissue simulant and porcine liver samples, respectively. This research demonstrates the efficacy of this sensing method and the associated sensor system for detecting the occurrence of tissue slip events during surgical grasping and retraction.

## 1. Introduction

Although robotic surgical devices have helped improve surgical outcomes in minimally invasive surgery [1], their limited force control remains an issue, especially when manipulating soft biological tissues [1,2]. The lack of haptic feedback due to a physical separation between surgeon and patient can result in tissue trauma due to the over application of gripping forces or tissue slip due to insufficient grasping [3,4].

One of the main solutions identified in the literature to limit tissue trauma caused by excessive grasping forces is the inclusion of force sensors and haptic or visual feedback of the grasping force information to the surgeon. Although this method has been shown to reduce tissue trauma [5,6,7], it requires constant monitoring by the surgeon, increasing their cognitive load [8]. Furthermore, it only provides information on the forces that are being applied and not whether the tissue is being held securely. A more direct method for controlling grasping forces is through the identification and early detection of slip events. If the onset or a precursor phenomenon to the slip can be identified, then the clamp force can be adjusted automatically to prevent further slips. This could allow the surgical robot’s grasper to use minimal force to maintain a stable grasp of the tissue, helping to reduce tissue trauma while still allowing for reliable and controlled manipulation of the tissue.

Research into grasper systems has identified several that aim to detect or predict the occurrence of tissue slip as the basis for an automated surgical grasper. Khadem et al. [9] developed an automated surgical grasper that aimed to predict when a slip was going to occur based on prior experimentation, adjusting the clamping force relative to the current retraction force to stay within a pre-defined safe grasping zone, limiting its application to known and quantified tissues. Other systems have been developed to detect when a slip is occurring, rather than attempting to predict it. The most mature of these is that developed by Burkhard et al. [10]; this sensor uses a technique based on hot-wire anemometry to monitor the variation in the heat flux as an indicator of slip. This method has been demonstrated to successfully detect slips for a range of biological tissues [10] and has been utilised for the grasping automation of surgical instruments [8]. Another instrumented grasper employed force sensors to detect the onset of slip by monitoring the normal and shear force at the face of the grasper and using these to calculate when the coefficient of friction first peaks to indicate that slip is occurring [11].

Both of the slip detection sensors described above utilise a single sensing node and thus cannot identify localised slip behaviour across the grasper. As a result, they are better suited to the detection of larger macro slips. Macro slips occur when the global shear force exceeds the global frictional force across the contact, leading to the entirety of the contact slipping [12]. The use of more localised sensing methods would allow for the detection of *incipient slips* [13], which are small localised slip phenomena that occur when shear forces in a localised region exceed the corresponding frictional forces, causing local slip while the remainder of the contact remains held securely [12]. As the shear force increases, the number of these incipient slip events will increase until the global shear force exceeds the global frictional force, at which point the surface enters a state of macro slip, where the whole of the contact area is slipping freely.

A key aspect of being able to detect incipient slip is being able to preferentially encourage it to occur in a predictable and repeatable manner [13]. This strategy utilises the human finger and employs a curved surface to create a normal force distribution, resulting in a frictional force distribution, which leads to incipient slip occurring towards the outer edge of the finger pad while the middle finger continues to grip securely [14]. This method of utilising the variation in normal and thus frictional force distribution to encourage preferential incipient slip has been utilised significantly by the wider robotics community [13,15,16] but has not yet been applied in the field of surgical grasping. An in-depth review of incipient slip sensors can be found in this recent review [13].

Other than our prior work, the only reported example of a sensor designed to detect incipient slip in surgery employs an approach that exploits the deformable nature of human tissue and monitors the change in material stiffness to estimate the first stages of macro slip [17]. However, this system only monitors the global shear forces rather than trying to identify localised slips, resulting in only a short time window for mitigating action to be taken [17].

The aim of our research is to produce an instrumented surgical grasper capable of inducing and then detecting preferential incipient slip before macro slip occurs that is applicable for a range of soft biological tissues and would be compatible with current surgical grasper designs. This sensor has already been demonstrated for the successful automation of surgical grasping using tissue simulants [18]. In this paper, we present the sensor concept and design before analysing its ability to reliably detect incipient slip over a wide range of test conditions, representative of those used in surgery. Initial experiments were conducted using a range of tissue simulants to define the slip detection algorithm that was utilised in [18] and to evaluate it on a repeatable substrate. The complete system was then applied to the grasping of porcine liver tissue to demonstrate its efficacy for real-world application in surgical grasping for the early detection of slip when manipulating soft biological tissues.

This work builds on our prior work in the area [18,19,20]. Here, our contribution is to rigorously evaluate the ability of the system to detect slip under a wider range of clearly defined test conditions for force, retraction speed, and material stiffness. In this paper, we also extend the testing to evaluate the system using porcine liver samples, providing a more representative test of the system for use in the grasping of soft biological tissues.

## 2. System Development

### 2.1. Concept and Requirements

The foundation of this sensing approach is to induce localised incipient slip in a predictable and repeatable manner so that it can then be detected. To achieve this, we utilised biomimicry of the human finger [14], similar to other robotic graspers [13,15,16]. A convex curved grasper face was used to create areas of high normal and thus frictional force in the middle of the grasper, with these forces gradually decreasing towards the edges of the curved face, promoting tissue to slip first at the outer sections of the grasper [18,19] (Figure 1). By separating the grasper into a series of independently movable ‘islands’ and then tracking the displacement of these islands during tissue retraction, it was possible to detect when the edge sections started to slip relative to the middle of the grasper. In the initial phase of a typical tissue retraction process, the shear forces will initially be low; therefore, friction forces are dominant across the grasper face and all of the islands will grip the tissue securely. As retraction progresses, the shear force increases and in the areas of low normal and frictional force (towards the outer edge of the grasper), the local shear force will exceed the local frictional force, resulting in a localised (incipient) slip, whereas the middle will continue to maintain a stable grip on the tissue and move with it (Figure 1). This results in a differential in the relative displacement between the islands at the edge and middle of the grasper, which can then be used as an indicator of the presence of incipient slip of the outer islands.

### 2.2. System Design and Fabrication

A scaled model of a surgical grasper was created to evaluate the sensor concept. A curved grasper face (r = 100.25 mm) was separated into a 5 × 3 grid of movable islands across the width and length of the grasper (Figure 2). The separation along the length, axial to the curvature of the face, was to try to isolate the slip effects across the width due to the normal force variation from those along the length of the grasper caused by the deformable properties of the tissue [19,20].

Each island consisted of a 3D-printed rigid upper gripping surface (Rigid 4000 Resin, Formlabs, Somerville, MA, USA) to securely hold the tissue, with a 1 mm-thick layer of silicone elastomer (Ecoflex 00-30, Smooth-on, Macungie, PA, USA) placed below to allow each island to move freely, allowing for displacement differentials to occur between them (Figure 2). The upper gripping surface was patterned with hexagonal features (0.75 mm width, height, and separation) to provide an isometric frictional performance that was suitable for the gripping of soft lubricated biological tissues [19,21].

To monitor the displacement differential between the outer and middle islands, neodymium disc magnets (2 mm diameter × 0.5 mm thickness) were embedded in the base of the rigid upper gripping surface of the front-left, middle, and right islands (Figure 2). A tri-axis Hall-effect sensor (MLX90393, Melexis, Ypres, Belgium) was then placed below each of these islands to track their displacement by monitoring the movement of the magnetic field (B${}_{x,y,z}$) that occurred as the magnet moved above the sensor chip when a force (F${}_{x,y,z}$) was applied to the upper gripping surface, and the elastomer layer below deformed (Figure 2). The magnetic fields (B${}_{x,y,z}$) could then be converted into a corresponding displacement based on previously calculated sensor calibrations. These sensors were selected based on prior work, which indicated that they could provide sufficient sensitivity for the detection of the slip differentials that occur between the outer edge and middle of the grasper, with a sufficiently compact footprint to fit three nodes across the width of the grasper face [19,22]; the thermal effects are considered negligible due to the minimal temperature variation within the human body. These sensors were configured to sample the magnetic field at a frequency of 408 Hz.

### 2.3. Signal Processing

To determine the relationship between the magnetic field and island displacement for each sensor node, a custom three-axis sensor calibration system was constructed. This was used to sweep a magnet in a 3D volume (covering the x-, y-, and z-axes) above the Hall-effect sensor, with the magnet connected directly to the linear stage assembly to ensure that there was no slip, thus ensuring that the stage displacement corresponded to the resultant magnetic flux. Full details of the instrumentation and process are detailed in [22]). The sweep in the x-y plane was conducted from −2 to 2 mm at 0.2 mm/s for each step in z (−0.65:1.25 mm in 0.1 mm increments), where {0,0,0} was the position of the neodymium magnet centre when the island was unloaded and centred over the Hall-effect sensor. A neural network (Matlab, Mathworks, Natick, MA, USA) was then trained to fit the magnetic field reading of the sensor to the displacements measured by the calibration system. This neural network utilised a two-layer feedforward network [22] with 40 neurons in the hidden layer and was trained using a Bayesian regularization backpropagation algorithm due to the nonlinear relationship between the displacement and the magnetic field [22]. This neural network provided a strong correlation with the validation data, with root mean squared errors of 0.029 mm, 0.025 mm, and 0.018 mm, in the x-, y-, and z-axes, respectively. The output of the neural network was post-processed using a third-order Butterworth filter with a cutoff frequency of 10 Hz to attenuate high-frequency noise.

### 2.4. Slip Detection

In the context of this sensing system, an incipient slip is defined as the occurrence of a slip between the grasped tissue and at least one of the grasper islands while the other islands retain a stable grasp of the tissue. When using a convex curved grasper (see Figure 1), the incipient slip is expected to occur first at the outer islands of the grasper. Accordingly, the tracking island displacement and considering the differential between the outer and inner islands provides a means of identifying when the incipient slip starts to occur at the grasper face. An algorithm was developed based on the results gathered using three different tissue simulants over a wide range of applied clamping loads and retraction speeds to automatically identify the onset of an incipient slip.

A typical response from this sensor system retracting a tissue simulant is presented in Figure 3, which shows the variation in the displacement of the front-left, middle, and right islands throughout the full retraction. A similar response was observed across the full range of test conditions investigated for the tissue simulants. During the initial stages of the retraction when the shear forces were low, the outer and middle islands moved together with the retracting tissue at the same velocity. However, as the shear force increased with further retractions, the outer islands started to slip against the tissue due to the lower normal and frictional forces at these points. This resulted in the decreasing velocity of these islands as they slipped more and more against the tissue, indicated by the plateauing of the island’s displacement, whereas the middle island continued to grip the tissue securely and move with it. Therefore, by comparing the velocities of the left ($V_{l}$) and right ($V_{r}$) outer islands to that of the middle island ($V_{m}$), it was possible to define the magnitude of the relative slip differential between them. The ratio of the velocity of the middle to the outer islands is termed the slip ratio (ϕ); this algorithm has previously been used by us for the automation of the force control during the grasping and retraction of tissue simulants [18].
(1)$$\phi =\min\left[\frac{V_{l}}{V_{m}},\frac{V_{r}}{V_{m}}\right]$$

If the velocity of the middle islands was positive, then a slip ratio (ϕ) = 1 indicated that the outer and middle islands were moving at the same velocity; therefore, there was no slip occurring at the outer islands relative to the middle. However, as the slip ratio (ϕ) decreased, the relative velocity of the outer islands decreased due to an increasing amount of slip occurring between these islands and the tissue simulant. When the slip ratio reached 0, the outer islands were no longer moving with the tissue simulant as it retracted, indicating that the tissue was freely slipping against the outer island(s). Over the full range of parameters investigated (three tissue simulants, three clamp loads, three retraction speeds), a ϕ value of 0.2 or less was found to be a reliable and robust indicator that the outer islands were encountering a significant amount of slip relative to the middle, without producing a false early indicator of slip, or resulting in slip being detected too late for any mitigating action to be taken. This algorithm was activated only once the middle island had moved 0.02 mm in the direction of the shear to avoid false detection due to signal noise (i.e., the retraction process must have started).

## 3. Materials and Methods

### 3.1. Experimental Set Up

A test rig was developed, which simulated the grasping and retraction of tissue that occurs during robotic surgery, as shown in Figure 4, to evaluate the efficacy of the sensor in detecting incipient slip events. The instrumented grasper face was attached to a pneumatic piston (MGPM20TF-75Z, SMC, Tokyo, Japan) to grasp the tissue. The grasping force was controlled via a pneumatic regulator (ITV1030, SMC, Tokyo, Japan), which had been pre-calibrated against a reference load cell. A linear load tester (Instron 5940, Instron, Norwood, MA, USA) with a pneumatic jaw was employed to simulate the retraction motion of the grasper. The global shear force and retraction speed were monitored via the linear load tester and the array of Hall-effect sensors in the grasper were read via a microcontroller (Teensy 3.6, PJRC, Portland, OR, USA). A real-time embedded controller (MyRIO, National Instruments, Austin, TX, USA) was then used to synchronise and interface these systems to control the clamping and retraction motions and record the data at a frequency of 100 Hz. In addition, a camera (AVE2, Instron, Norwood, MA, USA) was positioned behind the clear acrylic counterface to record the tissue movement at a frequency of 50 Hz.

### 3.2. Experiment I: Silicone Tissue Simulants

An initial assessment of the sensor system and slip detection method was conducted to understand the sensor’s response to slip events and the underlying slip mechanics. This was conducted using silicone tissue simulant to emulate the properties of soft tissues and enable controlled and repeatable testing.

Three tissue simulants with different material tensile stiffness were used: Mat A (*E* = 241 kPa), Mat B (*E* = 320 kPa), and Mat C (*E* = 610 kPa). Each tissue simulant comprised three layers of silicone elastomer (Ecoflex 00-30, Smooth-on), encapsulating internal layers of strain-limiting deformable spandex fabric located 0.3 mm below the upper and lower tissue surfaces. The fabric layers were altered to vary tensile stiffness (as detailed above) while maintaining consistent frictional and compressive characteristics (1051 ± 60 kPa). After fabrication, the simulants were laser-cut into 100 × 20 × 3 mm test samples. A speckled pattern was applied on one face using enamel spray paint to allow the displacement to be tracked using Digital Image Correlation (DIC) (GoM Correlate, GoM). A layer of surfactant lubricant was applied immediately prior to the test to mimic the serous fluid coating commonly exhibited on soft tissues [23]. The material stiffness of each sample was measured using the ASTM D412 Type C tensile method, and their response was found to be similar to that of liver tissue [24].

Testing consisted of a simulated grasp process, exploring the factors of the grasp load and retraction speed. In each test, a fixed grasping load was applied to the tissue simulant and was then retracted at a constant speed for 30 mm. Grasping loads of 10 N, 20 N, and 30 N were evaluated and these were identified as being representative of the grasping pressures observed during surgical practice [25,26]. Retraction speeds of 1, 2, and 5 mm/s were selected based on those typically used for tissue manipulation [27]. Five repeats were conducted for each test case.

### 3.3. Experiment II: Porcine Liver

To evaluate the performance and characteristics of the sensor system in a more surgically realistic configuration, a series of tests were conducted using ex vivo porcine liver samples.

In these tests, the same sensor and grasper design, as well as the slip detection algorithm developed for the tissue simulants, was used to allow direct comparison with the tissue simulants. The tests explored the variation in the clamp force, using the same configuration of 10, 20, and 30 N, with a fixed retraction speed of 2 mm/s for all tests, as this was the average speed used during the manipulation tasks [27], and experiments with the tissue simulants indicated that this speed had a very predictable and quantifiable influence on tissue slip and the rate of slip propagation. For each load condition, two separate porcine livers were analysed, with 5 test samples taken from each liver for a total of 10 tests at each load condition. To prepare the samples, the livers were sliced into thin strips with a nominal thickness varying from 4 to 12 mm. The variability was the result of the soft and deformable nature of the liver tissue making sample preparation challenging (in contrast to the high tolerances achieved with the fabricated simulants); however, this provided a representative reflection of the conditions expected in a surgical environment. A nominal 100 × 20 mm rectangle was cut from each slice using a stencil and a scalpel and was then immersed in a saline solution to prevent drying, providing a more representative sample [10]. For the characterisation of the material properties of each liver, three ASTM D412 Type C tensile specimens and one 100 mm × 20 mm compression specimen were cut from each liver. The tensile moduli ranged from 482 to 1304 kPa, with an average of 718 ± 223 kPa, which was comparable to the results seen in the literature [24]. Some test pieces were excluded due to artefacts in the sample (e.g., tears/holes in the tissue, material inhomogeneities) causing anomalous results. The average compressive stiffness was 875 ± 159 kPa; however, during the first 10% of compressive strain, it was significantly less than this, in the range of 100–300 kPa, which was similar to prior literature [28].

Each liver was retracted in the same manner as the tissue simulants; the specified clamp force was applied, followed by a 30 mm retraction of the tissue at 2 mm/s. Only a single repeat was conducted on each liver sample to mitigate the effects of tissue damage caused during the grasping and retraction process impacting on the response.

## 4. Results

### 4.1. Experiment I: Silicone Tissue Simulants

A summary of the representative results for the tissue simulant testing is provided in Figure 5a, which shows the change in displacement of the sensor islands at the front left, middle and right of the grasper throughout the retraction as a function of the material stiffness and applied clamping force. In all test cases, the movement of the left and right sensor islands plateaued in advance of the middle islands, indicating that the left and right islands were preferentially slipping as anticipated despite the significant variations in both the clamp load and material stiffness. The horizontal arrows on the graphs indicate the time difference between the detection of incipient slip and the occurrence of macro slip, termed the mitigation time ($\Delta t_{mitigation}$), and the time differential between the detection of incipient slip at one of the front-outer islands and the front-middle island ($\Delta t_{front}$). The changes in the material stiffness had a significant influence on the available mitigation time, as did the load, though these effects are better summarised in Figure 6.

Figure 5b displays the effect of retraction speed on sensor performance, showing the movement of the respective islands with respect to the displacement of the linear stage retracting the tissue in order to normalise the results along the x-axis for the different retraction speeds. From these results, it is evident that the various stages of slip occurred at similar levels of retraction (e.g., movement of the linear stage) despite the changes in the retraction speed.

A summary of the effects of the variation in the material stiffness, clamping force, and retraction speed is provided in Figure 6. These graphs compare the ways the different variables affect $\Delta t_{front}$ and $\Delta t_{mitigation}$. For tissue simulants Mat A and Mat C, the variation in the clamping force resulted in no significant change to the mitigation time. However, the increase in the clamp force from 10 N to 20 N did produce a significant increase in $\Delta t_{front}$ for these two materials (Figure 6a). For Mat B, an increase in the clamping force resulted in an increase in both $\Delta t_{front}$ and $\Delta t_{mitigation}$. The effects of the retraction speed appear to be inversely proportional to $\Delta t_{front}$ and $\Delta t_{mitigation}$, with a doubling in the retraction speed resulting in a 49% and 47% reduction in the time difference, respectively.

Figure 7 shows the representative results of the tissue simulant displacement and deformation during testing, as measured using DIC for the 20 N, 2 mm/s retraction cases for Mat A and Mat C. These data show that the displacement of the simulant varied between the front and rear of the grasper during the retraction, providing an indication of the magnitude of the displacement that occurred when incipient slip was first detected. For Mat A, when incipient slip was first detected at the outer-front islands, the simulant at the rear of the grasper moved only ca. 0.07 mm in comparison to ca. 0.49 mm at the front. For the stiffer material, Mat C, slip was detected when there was approximately 0.27 mm and 0.65 mm of displacement at the front and rear, respectively. It should be noted that these measurements were taken from the face of the tissue simulant contacting the smooth acrylic counterface, at which there was lower friction than at the grasper face. Although this resulted in more slip than occurred at the grasper contact, it provided a valuable indication of the overall characteristics of the tissue deformation.

### 4.2. Experiment II: Porcine Liver

The ex vivo testing with liver tissue was completed successfully, with no occurrence of tissue failure prior to gross slip occurring. Figure 8 provides a representative summary of the results for the tests in which incipient slip of the outer islands was successfully detected for the three different clamp-force conditions investigated. Across the full set of tests, incipient slip was reliably identified in 77% of cases, with detection rates of 80%, 80%, and 70% for the 10 N, 20 N, and 30 N test conditions, respectively.

As expected, in comparison to the tissue simulant experiment, there were significant variations in the results from the ex vivo testing across the different liver samples. This was due to the magnitude of the variations in the samples in terms of thickness, material properties, and the presence of anomalies (e.g., non-homogenous tissue containing features such as blood vessels). This variability was responsible for a range of phenomena, which resulted in unsuccessful attempts to detect incipient slip, as illustrated in Figure 9. Out of the unsuccessful defections, 71.4% were the result of an uneven compression or load application across the width of the grasper, and the simultaneous slip of all three islands (14.3%) and the lag of one of the sensors that caused the premature detection of slip (14.3%) were responsible for the remaining failed detections. The occurrence of mechanical snagging, where tissue catches on the islands and then releases rapidly, was seen in two cases but the manner in which it occurred did not affect the sensor’s ability to reliably detect slip.

A summary of how the variation in the clamp force affected the time between the detection of incipient slip at the front-outer and middle grasper islands, $\Delta t_{front}$, as well as the available mitigation time, $\Delta t_{mitigation}$, is shown in Figure 10. These values were only calculated for the tests in which slip detection was considered successful. There were no significant differences (*p* < 0.05) between any of the load conditions investigated; this is likely due to the high level of variability among the liver samples, which masked any potential trends.

## 5. Discussion

The results presented in this work demonstrate that measuring preferentially induced incipient slip as a means of detecting and preventing gross slip is an effective approach with relevance to surgery. When grasping tissue simulants, it was able to successfully detect slip in 100% of cases despite the large variations in the test conditions (Figure 5). For the more challenging case of ex vivo porcine liver samples, slip was accurately detected in 77% of cases despite still using the same algorithm and sensor developed for the silicone tissue simulants. The decrease in successful detections was due to the variability in the liver tissue samples compared to the silicone simulants.

The majority of slip detection failures that occurred when grasping the porcine liver samples were the result of unevenness in the magnitude of the tissue compression across the width of the grasper, which led to poor mechanical engagement with at least one of the outer sensor islands. This was usually due to either the variation in the liver sample thickness across the width of the grasper or was the result of anomalies within the tissue, e.g., holes, pits, or tissue variation (fat/liver tissue), which caused unexpected variations in the normal force. In some cases, this resulted in an asymmetrical slip that did not affect the performance of the sensor significantly, but in other cases, there was almost no engagement between one of the outer sensor islands and the liver tissue (Figure 9), resulting in these islands slipping instantly when the retraction started. However, it maybe possible to mitigate this issue by monitoring the normal force exerted on each island during the initial grasping and using this to determine the level of mechanical engagement of the different islands. The system could then determine the islands that should be monitored for the detection of incipient slip events. It is also possible that a smaller grasper face would be less susceptible to anomalies and thickness variations in the tissue, though further investigation is required to confirm this.

In addition to the uneven load distribution, there are three further abnormal results that are presented in Figure 9. The ‘snagging’ issue occurred as a result of tissue artefacts, such as holes or tears in the tissue catching on the sensor islands, leading to a catch and release of the tissue and associated spikes in sensor displacement in contrast to the more gradual changes that occurred during a grip dominated by friction. For the other two issues identified Figure 9, the occurrence of simultaneous slip and the lag in displacement of one of the sensor islands, based on the video footage of the slip, it is clear that the sensor was reporting the same motions that were occurring at the tissue–grasper interface, though the root cause of this behaviour has not yet been identified.

Although the sensor system was able to successfully detect slip for the tissue simulants in 100% of cases, across the full range of clamp forces, retraction speeds, and material stiffnesses, there were still indications of potential limitations to the approach. An evaluation of the effects of the variation in the clamp force for the tissue simulants (Figure 5a) indicated that at a low clamping force (10 N), there was little displacement of the left and right sensor islands before they started to slip. This suggests that there was low mechanical engagement between these outer islands and the tissue simulant due to the lower normal force. As the clamping force increased, the mechanical engagement also increased, and this can be observed by analysing $\Delta t_{front}$, the time between slip at the front-outer and middle grasper islands for Mat A and Mat C (Figure 6). When moving from 10 to 20 N, $\Delta t_{front}$ increased but then there were no further increases when the load increased to 30 N, suggesting that there was a minimum clamp force and level of mechanical engagement required to ensure that the sensor islands could reliably engage with the tissue. The grasper design could be modified to improve performance by altering the curvature of the grasper and/or the stiffness of the movable silicone below the upper gripping surface to suit the desired application and operating range. Mat B did not present the same behaviour to the variations in the clamp load, and this was suspected to be due to the different manner in which the restraining fabrics in this simulant were layered compared to Mat A and Mat C; the differing performance of Mat B was also observed in previous experiments [20].

For the porcine liver tissue, the variation in the clamp force did not appear to significantly affect the $\Delta t_{front}$ (Figure 10); this was likely due to the higher tissue thickness and low compressive stiffness, which allowed for significant mechanical engagement between the liver sample and the grasper islands over all three force conditions.

There was significant variation in the available mitigation time, $\Delta t_{mitigation}$, for the different tissue simulants. This was due to the variations in the rate of slip propagation between the front and rear of the grasper, with stiffer materials having a higher rate of slip propagation, resulting in a shorter time available for mitigating action to be taken before macro slip occurred [20]. However, the variations in the material stiffness of the tissue simulants appeared to have no significant effect on the sensor’s ability to detect slip of the edge islands relative to the middle (Figure 5a) or on $\Delta t_{front}$ for Mat A and Mat C across the full range of forces and retraction speeds investigated (Figure 6). This indicates that this slip detection method is at least partly independent of the tensile material properties of the grasped tissue, a highly desirable property for future deployment in surgical environments.

The long-term aim of this work is the application of the sensing system to a surgical robot. Making this viable requires addressing a number of challenges, recognised as limitations in the current system:Grasper size: The instrumented grasper presented in this paper is approximately a factor of 2.5 times the size of a standard robotic surgical grasper. This difference in scale is not expected to effect the fundamental slip mechanics that form the basis of this sensing method. The variation in the normal force to induce incipient slip has previously been utilised for a range of conventional robotic grippers [13], and the effect has even been observed to occur at the nanoscale asperity level of contacts [12], indicating that this phenomenon can be considered independent of the scale of the contact in this application. However, scale remains an issue with respect to manufacturing. Recent advances in Hall-effect sensor development will aid the development of more compact systems in the future [29].Tissue properties: The current sensor system and method have been demonstrated to be effective at detecting incipient slip of porcine liver samples despite the lack of optimisation of the sensor design or detection algorithm for this target. This indicates that the sensor system has some robustness to the material properties of the grasped tissue. However, further evaluation is now required to investigate the full range of tissues and mechanical properties for which this technique is applicable including variations in the size and shape of the tissue being manipulated.Surgical application: The grasping and retraction actions reported in this paper are necessarily simplified in comparison to the manipulation movements that occur during actual robotic surgery. In a surgical environment, retraction can include aspects of lateral and rotational movement. Standard surgical graspers also utilise a scissor-like mechanism to grasp tissue rather than the parallel action utilised here. Further experimentation using a simulated surgical system is planned to evaluate how these aspects can effect the performance and robustness of this sensing approach.

## 6. Conclusions and Further Work

In summary, the slip sensing approach presented in this work has been demonstrated as an effective way to detect the occurrence of incipient slip events for a variety of deformable tissue-like materials, including porcine liver. The sensor shows robustness to a variation in the tensile stiffness of the material, though there are indications that tissue compressibility and clamp load are factors that will affect the working range. A significant challenge to address in the future transition towards surgical use is the variability and anomalies present within biological tissue that can result in detection failures. Solutions have been identified to resolve these and, in general, this work shows the potential use of this method to automate the surgical grasping of soft biological tissue. The focus of future work is to scale down the sensor so that it can be integrated into a standard robotic surgical system and to optimise the detection algorithm to identify and mitigate for tissue anomalies and variability.

## Figures and Tables

**Figure 1 sensors-22-07956-f001:**
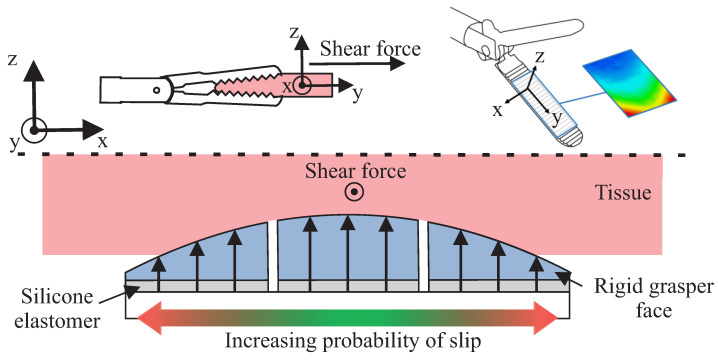
Diagram demonstrating the concept of encouraging incipient slips across the grasper face by varying the normal force using a curved surface.

**Figure 2 sensors-22-07956-f002:**
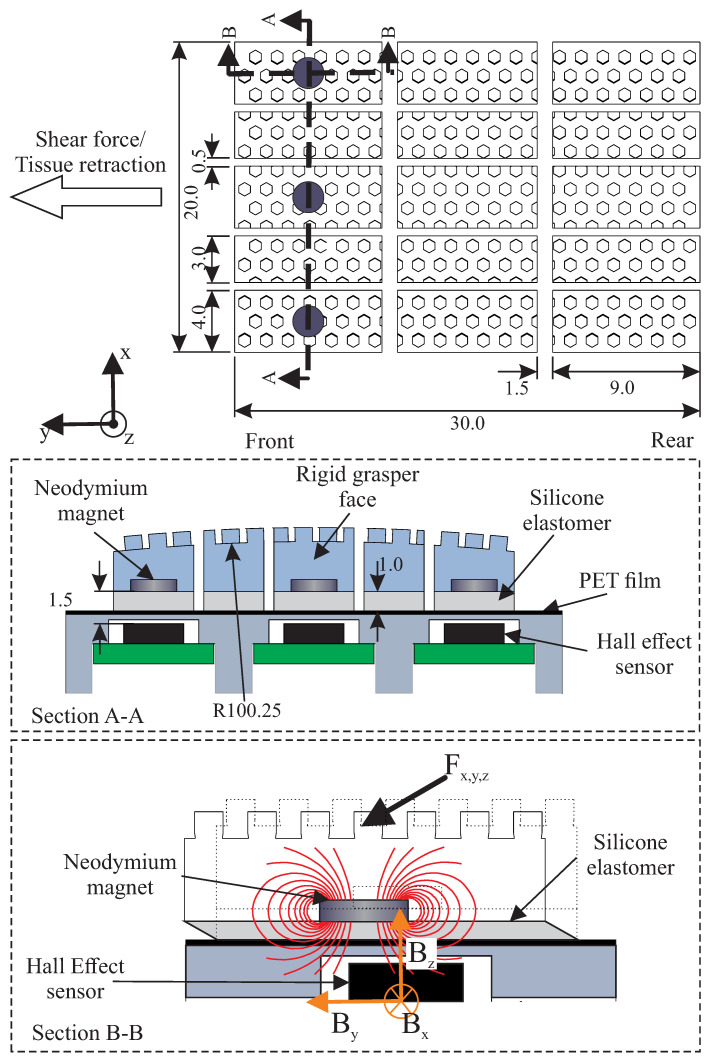
Detailed drawing of grasper design showing independently movable islands; the grey circles indicate the positions of the magnetic sensing nodes (all dimensions are in mm). Section A-A: Cross-section displaying the design of the sensored and passive islands. Section B-B: Cross-section showing how the magnetic field moves through the Hall-effect sensor as force is applied to the island.

**Figure 3 sensors-22-07956-f003:**
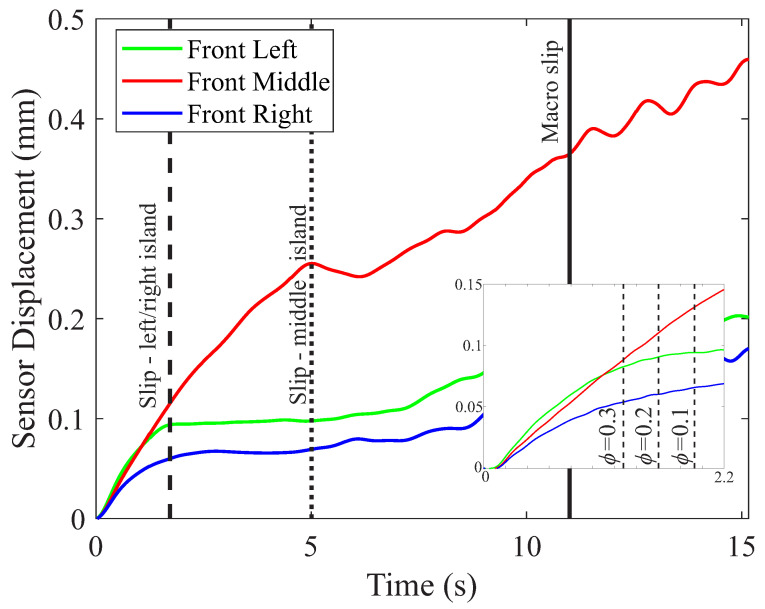
Example of typical displacement characteristics for the front-left, middle, and right islands of the grasper under a 20 N clamp load with a retraction speed of 2 mm/s (using Mat A). The inset shows how the variation in the slip ratio changes the time at which the incipient slip of the edge islands is detected.

**Figure 4 sensors-22-07956-f004:**
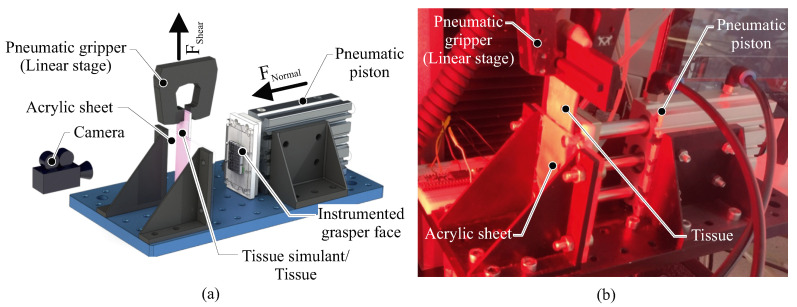
(**a**) A schematic showing the key components of the experimental system. (**b**) An image of the experimental system used for simulating surgical grasping and retraction.

**Figure 5 sensors-22-07956-f005:**
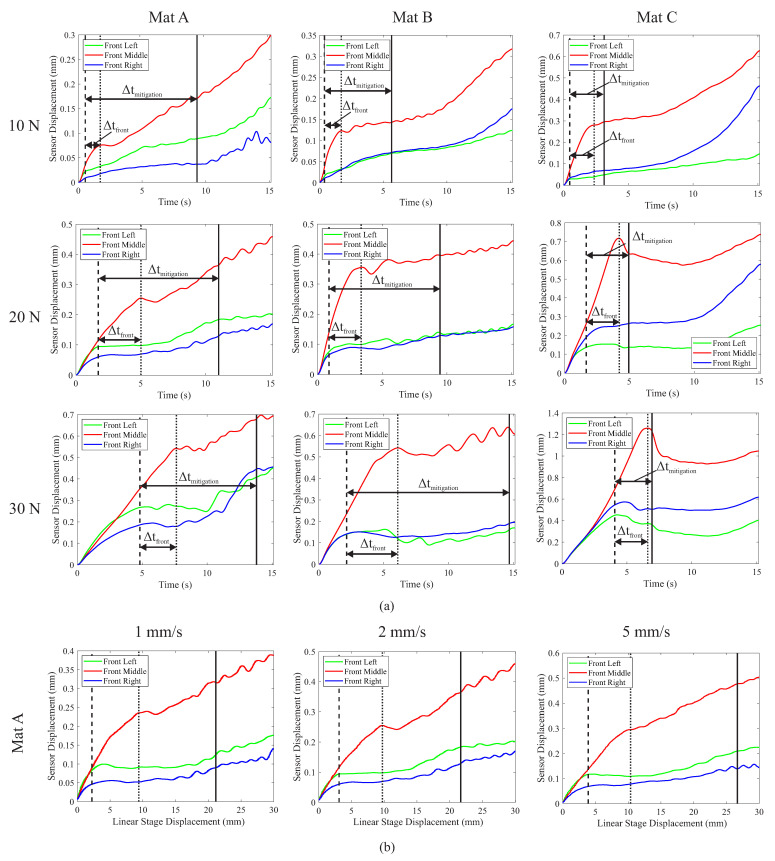
Graphs showing typical sensor displacements. The vertical dashed and dotted lines indicate the time slip was first detected at the outer and middle islands, respectively, whereas the solid vertical line indicates the time of macro slip. The horizontal arrows indicate the time difference between the detection of incipient slip and the occurrence of macro slip ($\Delta t_{mitigation}$), and the time between the detection of incipient slip of one of the front outer islands and the front middle island ($\Delta t_{front}$). (**a**) Sensor displacement vs. time for changes in force and material stiffness at a retraction speed of 2 mm/s. (**b**) Sensor displacement vs. linear stage displacement (retraction distance) for variations in retraction speed for Mat A under a 20 N clamping load.

**Figure 6 sensors-22-07956-f006:**
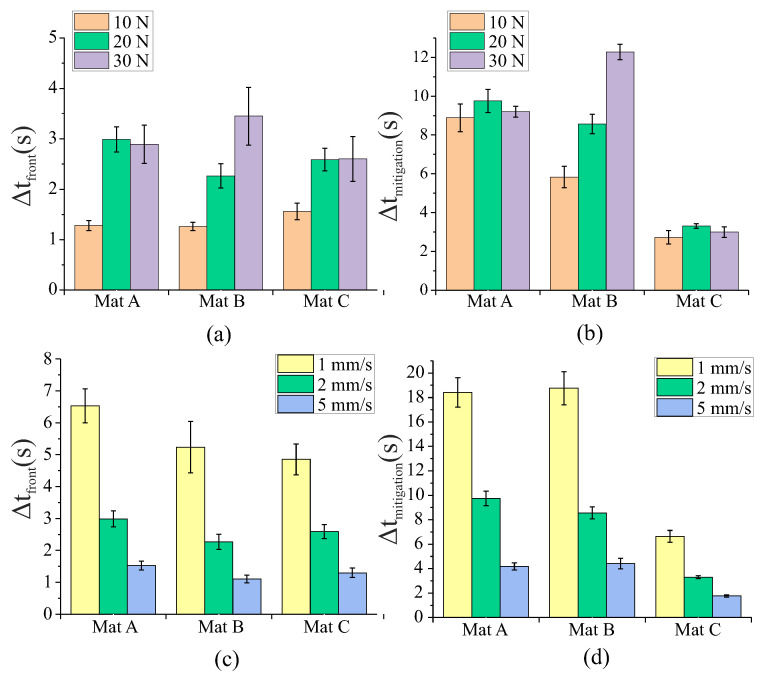
Graphs showing the mean time difference (N = 5) between (**a**) detection of slip at the outer and middle grasper islands ($\Delta t_{front}$) for a variation in the force and material stiffness with a 2 mm/s retraction speed, (**b**) available mitigation time ($\Delta t_{mitigation}$) for a variation in the force and material stiffness with a 2 mm/s retraction speed, (**c**) $\Delta t_{front}$ for a variation in the retraction speed and material stiffness with a 20 N clamp load, and (**d**) $\Delta t_{mitigation}$ for variations in the retraction speed and material stiffness with a 20 N clamp load. The error bars indicate the standard deviation.

**Figure 7 sensors-22-07956-f007:**
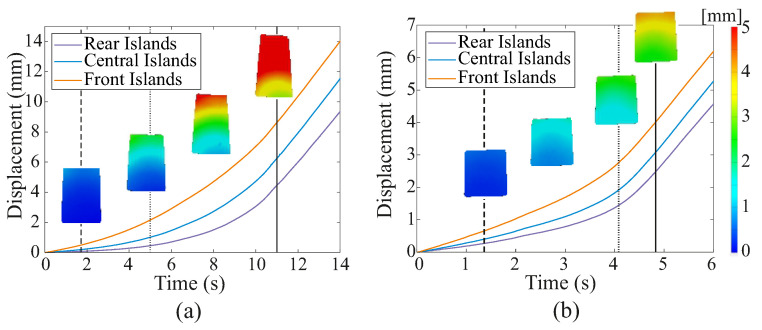
Typical results for average tissue displacement at the front, middle, and rear of the grasper face measured using DIC. The vertical dashed and dotted lines indicate that the time slip was first detected at the outer and middle islands, respectively, whereas the solid vertical line indicates the time of macro slip. The colour maps show the changes in the displacement profiles over time. (**a**) Mat A, 20 N, 2 mm/s. (**b**) Mat C, 20 N, 2 mm/s.

**Figure 8 sensors-22-07956-f008:**
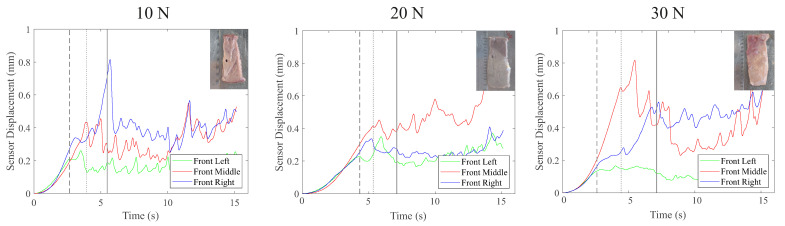
Graphs showing example sensor displacement for cases in which the early detection of slip was successful for porcine liver samples for various load cases. The vertical dashed and dotted lines indicate the time slip was first detected at the outer and middle islands, respectively, whereas the solid vertical line indicates the time of macro slip.

**Figure 9 sensors-22-07956-f009:**
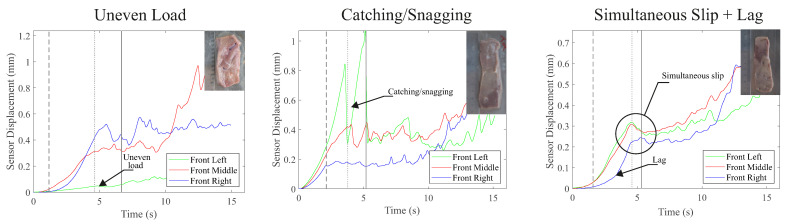
Graphs showing sensor displacements for cases in which early slip detection was unsuccessful or there were abnormal sensor responses for liver tissue retractions.

**Figure 10 sensors-22-07956-f010:**
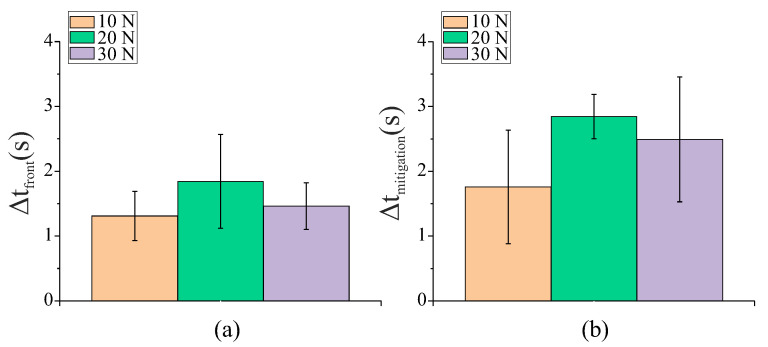
Graphs showing the mean time difference (N = 10) between the (**a**) detection of slip at the outer and middle grasper islands ($\Delta t_{front}$) and (**b**) available mitigation time ($\Delta t_{mitigation}$) for a variation in the clamp force with a 2 mm/s retraction speed. The error bars indicate the standard deviation.

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
