# Peer review of "Encouraging and Detecting Preferential Incipient Slip for Use in Slip Prevention in Robot-Assisted Surgery"

_sensors, 2022, doi:10.3390/s22207956_

Round 1
Reviewer 1 Report
1.
The authors should show sensor properties when the environment temperature is varied.
2.
It seems that this sensor is very large.
I don't understand the validity of your approach when the sensor becomes small.
3.
The authors should show a picture of the experiment environment systems.
Author Response
Dear Reviewer,
Thank you for your comments and suggestions to our manuscript. A full response is provided below; your comments are included italicised with our response beneath.
Comments:
- The authors should show sensor properties when the environment temperature is varied.
The intended environment for this sensor is within the human body in a surgical scenario. This environment exhibits minimal variation in temperature, with a nominal variation between 36.5–37.6 °C in a regulated surgical environment. The MLX90393 sensors selected have an operating range of -20C to 85C, with thermal drift of +/-3% at extremes of this range, so thermal variation was considered outside the scope of this study and overall a negligible factor in this particular application. However we understand that this consideration should be recognised and justified, accordingly we have added a short statement to this effect in the amended manuscript (137-138).
- It seems that this sensor is very large. I don't understand the validity of your approach when the sensor becomes small.
We recognise that the current prototype sensor is larger than a surgical grasper. However, the slip mechanics on which this system depends are independent of the scale of the contact. A wide variety of conventional robotic grippers have used this principle (the variation of normal force) to induce incipient slip, as reported in our introduction. This phenomenon has even been observed at the nanoscale asperity to asperity level of surface contacts.
To ensure these aspects are clear, we have included a limitations section at the end of the discussion to detail the major differences between our set up and that used in surgical practice (401-428).
- The authors should show a picture of the experiment environment systems.
We have included an image of the test set up used to simulate grasping and retraction of tissue in addition to the schematic diagram, shown in the revised manuscript (Fig.4).
Reviewer 2 Report
Further experiments with a real robot gripper will significantly improve in terms of research.
Author Response
Dear Reviewer,
Thank you for your suggestion to our research. We have responded to your comment below and highlighted updates in the revised manuscript.
Comments:
- Further experiments with a real robot gripper will significantly improve in terms of research.
We agree and have stated that this will form a core part of our future work. We have now also included a limitations section which discusses the various differences between our set up and a surgical robotic grasper (401-428). Full evaluation of the system within a robotic grasper was beyond the scope of this current work in which we are progressively working through Technology Readiness Levels (TRLs) toward clinical adoption. Our future plan is to integrate this sensor technology into a DVRK robotic surgical grasper for lab based ex vivo evaluation.
Reviewer 3 Report
This paper proposed a sensing method and system for early detection of slip events when a soft gripper is grabbing soft tissues by using machine learning. This method used in minimally invasive surgery as an early assessment helps to eliminate the probability of tissue trauma. In summary, this paper is well described and can be published in the journal of Sensors with the following question to be solved.
1. Is the grasper design Fig.1 commonly used in minimally invasive surgery? There are other shapes of grasper such as human fingers, which has more bending features that can prevent slippery better than your design.
2. What is the sensitivity of the data rate of the hall effect sensor?
3. The sweep rate of each axis is 0.2 mm/s, which means you need the 20s to finish one axis. So how do you make sure the sweep of the x-y plan happened at the same time especially when there is a slippery?
4. The total displacement shown in Fig.3 is around 0.5mm, while the time it cost is 15s. Is the real minimal invasive surgery owning the same rate? How do you compare the experiment result for practical use?
Author Response
Dear Reviewer,
Thank you for your comments and suggestions to our manuscript. A full response is provided below; your comments are included italicised with our response beneath.
Comments:
This paper proposed a sensing method and system for early detection of slip events when a soft gripper is grabbing soft tissues by using machine learning. This method used in minimally invasive surgery as an early assessment helps to eliminate the probability of tissue trauma. In summary, this paper is well described and can be published in the journal of Sensors with the following question to be solved.
Thank you for your appreciation of the quality of the work we have conducted, as well as the detailed comments you have provided to help improve the paper.
- Is the grasper design Fig.1 commonly used in minimally invasive surgery? There are other shapes of grasper such as human fingers, which has more bending features that can prevent slippery better than your design.
Yes, the work is based on a conventional grasper profile used in laparoscopy and robotic minimally invasive surgery. Hence, we have used this as the basis for our sensor platform. There is likely to be design improvements that can be employed to improve the overall gripping performance, but the aim of this work was to design a sensor system that would be compatible with current grasper designs. This is detailed in the manuscript (77-78).
- What is the sensitivity of the data rate of the hall effect sensor?
The Hall Effect sensor chips are set to monitor the magnetic field at a frequency of 408 Hz, though a reading is only taken by the microcontroller every 100 Hz. We have included this additional information in the manuscript (139).
- The sweep rate of each axis is 0.2 mm/s, which means you need the 20s to finish one axis. So how do you make sure the sweep of the x-y plan happened at the same time especially when there is a slippery?
The magnet is connected directly to the linear stages that are moving it above the Hall Effect sensor during the calibration procedure. This avoids the issue of slip during calibration and ensures that the measured displacements can be accurately correlated with those from the Hall Effect sensor measurements. We have added additional information to clarify this (144-146)
- The total displacement shown in Fig.3 is around 0.5mm, while the time it cost is 15s. Is the real minimal invasive surgery owning the same rate? How do you compare the experiment result for practical use?
The rates and loads selected were based on those used in minimally invasive surgery. This is stated in the methods section of the paper (231-236). The sensor displacement though will be much lower than that of the linear stage as the tissue being manipulated is deformable, therefore a significant amount of the displacement of the linear stage is taken up by the deformation and stretching of the tissue. However we have included additional information in the limitations sections which details the main differences between our experiment and conditions in-vivo (401-428).